# Dynamic and Steric Sea-level Changes due to a Collapsing AMOC in the Community Earth System Model

René M. van Westen<sup>1</sup>, Caroline A. Katsman<sup>1,2</sup>, and Dewi Le Bars<sup>1</sup>

Correspondence: René M. van Westen (r.m.vanwesten@uu.nl)

Abstract. A collapse of the Atlantic Meridional Overturning Circulation (AMOC) leads to a redistribution of dynamic sea level (DSL) across the global ocean surface. Here, we investigate AMOC-induced DSL and steric sea-level responses using the Community Earth System Model and two stand-alone ocean configurations (strongly eddying and parameterising eddy effects) with the Parallel Ocean Program. For our analysis, we employ various quasi-equilibrium freshwater hosing experiments in which AMOC collapses were reported. As the AMOC begins to collapse, the DSL substantially rises over the Atlantic Ocean and Arctic Ocean, with the largest DSL changes reaching 6 mm yr $^{-1}$  over the North Atlantic Ocean. In densely-populated coastal regions along the North Atlantic Ocean, DSL trends of up to 4 mm yr $^{-1}$  are found, potentially doubling local sea-level rise rates under an AMOC collapse scenario. Given the quasi-equilibrium approach, the hosing contribution to DSL trends is relatively small for periods of  $\leq 100$  years but becomes increasingly important over longer timescale. Moreover, an AMOC collapse increases the radiative imbalance at the top of the atmosphere up to +0.5 W m $^{-2}$ , with the excess heat being absorbed by the ocean, leading to more than 20 cm of global mean thermosteric sea-level rise. These results highlight the potential value of accounting for an AMOC collapse scenario when developing or applying sea-level rise projections for the North Atlantic Ocean.

#### 1 Introduction

A collapse of the Atlantic Meridional Overturning Circulation (AMOC) modifies the planetary heat and salinity redistribution and this causes large-scale climate shifts (Orihuela-Pinto et al., 2022). For example, the Northern Hemisphere cools and receives less precipitation under a reduced AMOC strength (Bellomo et al., 2023; van Westen et al., 2024b). Certain regions, such as Europe, are expected to see drastic changes in their present-day climate. The European climate would experience more intense winter storms and cold extremes, and more droughts (Vellinga and Wood, 2002; Jacob et al., 2005; Brayshaw et al., 2009; Jackson et al., 2015; Meccia et al., 2024; van Westen and Baatsen, 2025; van Westen et al., 2025b).

Apart from atmospheric impacts, the AMOC also modulates dynamic sea level (DSL) (Bryan, 1996; Levermann et al., 2005). DSL is the height of the sea surface above the geoid and has a global mean of zero (Gregory et al., 2019). The regional DSL is primarily determined by the background ocean circulation and density, both of which are influenced by different AMOC strengths (Vanderborght et al., 2025). The AMOC strength is expected to decline under future climate change (Weijer et al.,

<sup>&</sup>lt;sup>1</sup>Royal Netherlands Meteorological Institute, De Bilt, the Netherlands

<sup>&</sup>lt;sup>2</sup>Environmental Fluid Mechanics Section, Department of Hydraulic Engineering, Faculty of Civil Engineering and Geosciences, Delft University of Technology, Delft, The Netherlands

2020; Bonan et al., 2025), and there is also a risk of a complete AMOC collapse (van Westen et al., 2025d; Drijfhout et al., 2025). Global climate models project DSL rise over the North Atlantic Ocean (≥ 40°N) and the Arctic Ocean under AMOC weakening (Landerer et al., 2007; Katsman et al., 2008; Yin et al., 2009; Chen et al., 2019; Lyu et al., 2020), with local DSL trends exceeding +4 mm yr<sup>-1</sup> under a high-emission scenario over the 21<sup>st</sup> century (Ferrero et al., 2021; Pardaens, 2023). This DSL rise over the North Atlantic Ocean and Arctic Ocean becomes even larger when the AMOC fully collapses and can regionally reach up to +1 m (Levermann et al., 2005; van Westen et al., 2024b, 2025a). Evidence of AMOC variability has already been detected in sea-level observations from both satellite altimetry and tide gauges along North Atlantic coasts (Bingham and Hughes, 2009; Little et al., 2019). This is important because it indicates that AMOC fluctuations can influence flood risk, which is projected to increase under a weaker AMOC (Volkov et al., 2023; Howard et al., 2024).

The latest generation of coupled climate models only project significant (18 – 45%) AMOC weakening over the 21<sup>st</sup> century, and an AMOC collapse before 2100 is assessed as unlikely (Weijer et al., 2020; Fox-Kemper et al., 2021; Bonan et al., 2025). There are indications that most climate models have a too stable AMOC and likely underestimate the risk of an AMOC tipping event under climate change (Van Westen and Dijkstra, 2024; Vanderborght et al., 2025). If the AMOC would start to collapse this century (Ditlevsen and Ditlevsen, 2023; van Westen et al., 2025d; Drijfhout et al., 2025), the AMOC strength would reduce faster than in the latest Intergovernmental Panel for Climate Change (IPCC) assessment (Fox-Kemper et al., 2021). This means that DSLs over the North Atlantic Ocean and Arctic Ocean would increase faster than is currently anticipated for. This information on accelerated DSL rise is crucial for North Atlantic coastal communities and for developing adaptation strategies to sea-level rise (Haasnoot et al., 2018; Biesbroek et al., 2025).

Regional DSL projections are sensitive to the climate model mean state, model biases, and wind and buoyancy forcing (Lyu et al., 2020; Jesse et al., 2024). This complicates efforts to disentangle the individual contributions of 21<sup>st</sup> century AMOC weakening and climate change to DSL projections. The AMOC contribution to DSL changes can now be isolated using recent simulations performed with the Community Earth System Model (CESM, version 1.0.5), in which a slowly increasing freshwater flux forcing (i.e., hosing) causes an AMOC collapse (van Westen et al., 2024b). The AMOC tipping event is driven by intrinsic climate feedbacks, allowing DSL changes associated solely with the AMOC collapse to be isolated. Global mean steric sea-level changes caused by a collapsing AMOC can also be studied using the CESM. Inspired by the work of Levermann et al. (2005), here we aim to revisit DSL changes in a modern complex climate model under a collapsing AMOC.

The structure of this study is as follows. Section 2 introduces the CESM configuration, together with a description of the two stand-alone ocean simulations used. In Section 3, we present DSL changes under a collapsing AMOC and consider DSL changes for different AMOC mean states. Steric sea-level variations are examined in Section 4. Finally, Section 5 summarises and discusses the main findings.

#### 55 2 Methods

#### 2.1 Climate Model Simulations

For our analysis we will make use of the fully-coupled CESM as in van Westen et al. (2024b). This CESM version has horizontal resolutions of  $1^{\circ}$  for the ocean/sea ice and  $2^{\circ}$  for the atmosphere/land components, respectively. The CESM has constant pre-industrial greenhouse gas concentrations and was forced under an increasing surface freshwater flux forcing,  $F_H$ , which was applied over the latitude bands between  $20^{\circ}$ N and  $50^{\circ}$ N in the Atlantic Ocean. This freshwater flux forcing was compensated over the remaining parts of the ocean surface to conserve the total ocean salinity. The  $F_H$  was increased at a slow rate of  $3 \times 10^{-4}$  Sv yr $^{-1}$ , reaching a maximum value of  $F_H = 0.66$  Sv (model year 2,200). The slow forcing rate ensures that the AMOC remains close to its equilibrium for that particular  $F_H$  (van Westen et al., 2024a, 2025c) and that transitions are caused by internal (ocean) dynamics. The AMOC reaches its tipping point in model year 1,758 ( $F_H = 0.527$  Sv, van Westen et al. (2024b)) and takes about 100 years to collapse, the AMOC strength time series is shown in Figure 1a (black curve). Below, we refer to this quasi-equilibrium (QE) hosing simulation as the low-resolution CESM (LR-CESM).

**Figure 1.** (a): The AMOC strength at 1,000 m and 26°N for the quasi-equilibrium LR-CESM, HR-POP and LR-POP. (b – d): The time-mean DSL (first 50 model years) for the LR-CESM, HR-POP and LR-POP.

75

One side effect of this hosing approach is that DSL is directly influenced by variations in the freshwater flux forcing through density changes. This  $F_H$  contribution on DSL is small for short time intervals, and internal dynamics such as an AMOC collapse ( $\Delta F_H = 0.03$  Sv) then dominate DSL responses. However, over the full QE hosing simulation ( $\Delta F_H = 0.66$  Sv) this contribution needs to be considered and will be quantified by analysing the accompanying backward simulation that was performed. Starting in model year 2,200 ( $F_H = 0.66$  Sv) of the LR-CESM, the  $F_H$  was decreased at the same rate of  $3 \times 10^{-4}$  Sv yr<sup>-1</sup>, resulting in a 4,400-year long QE hysteresis simulation (see also Figure 5a). The AMOC starts to recover from model year 4091 ( $F_H = 0.093$  Sv), which is at a much lower  $F_H$  than the collapse and clearly demonstrating AMOC hysteresis behaviour under varying  $F_H$  (van Westen and Dijkstra, 2023).

The DSL changes for different AMOC strengths can be isolated by analysing the same  $F_H$  within the multi-stable regime, such that the imposed freshwater flux forcing is cancelled. The QE hosing simulation is mostly in a weak transient state and to obtain a climate state (almost) free of transient effects we analyse the statistical equilibria for fixed  $F_H$ , indicated by  $\overline{F_H}$ . These equilibria have time-invariant statistics and their radiative imbalance at top of atmosphere is almost zero (van Westen and Baatsen, 2025), meaning that natural climate variability is dominant. Four statistical equilibria were obtained by branching simulations from the QE LR-CESM within the multi-stable regime and fixed  $F_H$ , these simulations were integrated for 500 years during which the AMOC equilibrates (van Westen et al., 2024a), the last 50 model years are considered for the analyses. The statistical equilibria were obtained for the 'AMOC on' regime at  $\overline{F_H} = 0.18$  Sv (model year 600) and  $\overline{F_H} = 0.45$  Sv (model year 1,500), and similarly for the 'AMOC off' regime at  $\overline{F_H} = 0.18$  Sv (model year 3,800) and  $\overline{F_H} = 0.45$  Sv (model year 2,900). The hosing-corrected DSL changes that arise from different AMOC regimes can then be determined by comparing 'AMOC off' to 'AMOC on' for both  $\overline{F_H} = 0.18$  Sv and  $\overline{F_H} = 0.45$  Sv.

The ocean component in the LR-CESM has a nominal horizontal resolution of  $1^{\circ}$  and is too coarse to explicitly resolve mesoscale processes (Hallberg, 2013), such as ocean eddies, hence these processes are parameterised. However, mesoscale processes (i.e., the horizontal resolution used) do influence DSLs (van Westen and Dijkstra, 2021) and this can now be addressed using a hosing simulation with the high-resolution  $(0.1^{\circ})$  stand-alone ocean simulation that is strongly eddying (van Westen et al., 2025a). This simulation was performed with the ocean component of the CESM: the Parallel Ocean Program (POP, version 2, Dukowicz and Smith (1994)). The POP has a prescribed surface forcing consisting of a seasonally repeating atmosphere and river run-off fields (Weijer et al., 2012; Toom et al., 2014). The strongly-eddying POP version is referred to as the high-resolution POP (HR-POP) and is accompanied by a low-resolution POP (LR-POP) version with  $1^{\circ}$  horizontal resolution; the LR-POP has the same horizontal resolution as in the LR-CESM. The HR-POP and LR-POP are also forced under the slowly increasing  $F_H$  and with maximum  $F_H$  values of 0.18 Sv (model year 600) and 0.45 Sv (model year 1,500), respectively. In these simulations, the AMOC starts to collapse from model year 420 ( $F_H = 0.126$  Sv) in the HR-POP and from model year 1,044 ( $F_H = 0.313$  Sv) in the LR-POP (van Westen et al., 2025a), their AMOC strengths are also shown in Figure 1a. Due to computational constraints, only a forward QE hosing simulation was performed for the HR-POP and LR-POP, hence the side effects of hosing on DSLs can only be analysed for the LR-CESM.

110

120

More details on the AMOC properties, AMOC tipping time estimates, and AMOC-induced climate impacts in all of these climate model simulations were presented in previous work (van Westen and Dijkstra, 2023; van Westen et al., 2024b, a, 2025a; van Westen and Baatsen, 2025; van Westen et al., 2025b, d).

## 2.2 Analysed Model Output

The DSL ( $\zeta$ ) is defined as:

05 
$$\zeta(x,y,t) = \eta(x,y,t) - B(x,y,t) - G(x,y)$$
 (1)

where  $\eta$  is the sea surface height, B is the inverse barometer correction, and G the geoid (Gregory et al., 2019). In climate models, the effects of B and G are not included and G is directly provided as an output variable (variable name 'SSH' for the CESM). Note that the globally-averaged G is very close to zero in the ocean component of the CESM and we uniformly removed this residual from the DSL fields. The time-mean (first 50 model years) DSLs for the LR-CESM, HR-POP and LR-POP are shown in Figures 1b,c,d, respectively, and their overall DSL patterns and amplitude agree well.

The ocean component in the CESM is volume conserving due to the Boussinesq approximation and the steric sea-level contribution is determined from post-processing the model output (Greatbatch, 1994). The local steric sea level is defined as (Richter et al., 2013):

$$\eta_s = \int_{T}^{0} \frac{\rho_0 - \rho(T, S, P)}{\rho_0} dz \tag{2}$$

where  $\rho$  is the in-situ density and  $\rho_0 = 1028$  kg m<sup>-3</sup>. Variations in the globally-averaged  $\eta_s$  ( $\eta_s^g$ ) are mainly caused by oceanic temperature changes as salinity is conserved; this is also known as the global mean thermosteric sea-level change (Gregory et al., 2019).

The analysis of the model output is conducted at a monthly frequency, and the time series are subsequently converted to yearly averages. We used a linear fit to determine (local) trends in the yearly-averaged DSL. Some local DSL time series display non-linear behaviour once the AMOC starts to collapse while their DSL responses (i.e., increasing or decreasing) are consistent over time. Hence, we used a Mann-Kendall trend test (Hussain and Mahmud, 2019) to determine the significance of the DSL trends. For assessing the significance of DSL changes in the time-mean states, we used a two-sided Welch's t-test.

# 3 Results - Dynamic sea-level responses

In this result section, we explore DSL changes under varying freshwater flux forcing conditions. First in Section 3.1, we analyse the forward QE hosing simulations for the LR-CESM, HR-POP and LR-POP. Here we quantify DSL changes and trends under a collapsing AMOC. Next in Section 3.2, the effects of the applied freshwater flux forcing on DSLs are presented, where we analyse the accompanying backward QE hosing simulation for the LR-CESM.






## 3.1 Dynamic sea-level responses under a collapsing AMOC

We divide the ocean surface into five distinct regions and their spatial extents are shown in Figure 2a. We determined the spatially-averaged DSL over these regions, indicated as  $DSL_i$  with i representing the region. The time series of  $DSL_i$  are shown in Figures 2b,c,d for the LR-CESM, HR-POP and LR-POP, respectively. The DSL<sub>i</sub> changes are fairly linear up to the AMOC tipping event in the three simulations and, as the timing of the AMOC tipping event differs among the simulations, we determine DSL<sub>i</sub> trends for comparison. These trends are indicated here as  $\nu_i = \frac{\partial \mathrm{DSL}_i}{\partial t}$  (or similarly  $\nu_i = \frac{\partial \mathrm{DSL}_i}{\partial F_H}$ ), where only the part of DSL<sub>i</sub> time series prior to the AMOC tipping event is considered (dashed lines in Figures 2b,c,d). The magnitudes of  $\nu_i$ , expressed in units of cm per kyr (1 cm per kyr = 0.01 mm per year), are displayed in the legend of Figures 2b,c,d. The Arctic Ocean and North Atlantic Ocean regions have the largest  $\nu$  in all simulations. For the latter region, the hosing is directly applied over the latitude bands between 20°N to 50°N and increases DSLs through freshening. The background circulation (overturning, gyres and eddies) then carries the imposed freshwater flux forcing into the Arctic Ocean (van Westen et al., 2024b; Vanderborght et al., 2025). As the globally-averaged DSL is zero, the increasing DSLs over the North Atlantic Ocean and Arctic Ocean are compensated over the remaining ocean regions. The largest DSL drop is found over the Indo-Pacific Ocean region and is remarkably consistent ( $\nu \approx -5.5$  cm per kyr) among the simulations. This consistent DSL drop is attributed to the negative freshwater flux forcing to conserve ocean salinity. The magnitude of  $\nu$  over the two remaining regions (South Atlantic Ocean and Southern Ocean) is relatively small with different  $\nu$  signs. For example for DSL over the South Atlantic region, the DSL is increasing (LR-CESM), decreasing (HR-POP) or remains near zero (LR-POP). These intermodel differences in  $\nu$  are attributed to ocean dynamics, as the applied hosing is identical across the simulations.

The largest DSL changes are found during the AMOC collapse, with relatively large DSL rise over the Arctic Ocean and North Atlantic Ocean. These two regions show large-scale freshening as a consequence of the salt-advection feedback that destabilises the AMOC (van Westen et al., 2024b; Vanderborght et al., 2025). The DSL time series over Arctic Ocean and North Atlantic Ocean have a comparable trajectory to their AMOC strength time series (cf. Figure 1a). This relation is quantified in Figures 2e,f,g, which present the changes in AMOC strength against DSL<sub>i</sub> changes, and is indicated by  $\phi_i = \frac{\partial DSL_i}{\partial AMOC}$ . Similar as before, the quantify  $\phi_i$  is determined up to the AMOC tipping event, and the relation is extrapolated to cover the whole range of AMOC strength changes. The DSL-AMOC responses align well with the extrapolated  $\phi$  relations for the North Atlantic Ocean and Arctic Ocean. This indicates that those two DSL<sub>i</sub> are strongly influenced by AMOC strength and that larger AMOC strength variations lead to larger DSL<sub>i</sub> changes. Note that DSL<sub>i</sub> still changes once the AMOC has equilibrated (i.e.,  $\Delta DSL_i \neq 0$  and  $\Delta AMOC \approx 0$ ), which is then caused by  $F_H$  variations. The AMOC-induced DSL responses also explain why the HR-POP has the largest  $\nu$  for the Arctic Ocean (23.9 cm per kyr) and North Atlantic Ocean (17 cm per kyr), as its AMOC is the most sensitive (i.e.,  $\frac{\partial AMOC}{\partial t}$ ) prior to its collapse compared to the LR-CESM and LR-POP. For the three remaining regions, an apparent DSL-AMOC relation may be present which is likely explained by balancing effects (to have a globally-averaged DSL of zero).

The results in Figure 2 demonstrate that an AMOC collapse influences DSLs. To further quantify these AMOC-induced DSL responses, we determine the linear DSL trends over three 101-year windows: before, during and after the AMOC collapse

Figure 2. (a): Definition of the five different regions. The percentages indicate the fraction of the total ocean surface, the remaining 0.5% is attributed to (semi-)enclosed seas and lakes. (b – d): Spatially-averaged DSL over five different regions for the LR-CESM, HR-POP and LR-POP. The DSL time series are displayed as their differences to the first 50 years and are then smoothed through a 25-year running mean to reduce the variability. The vertical gray line marks the onset of the AMOC collapse. The DSL trends are determined from model year 1 up to the AMOC tipping event and given in the legend ( $\nu$ ). The inset shows the DSL differences between the last 50 model years and first 50 model years. (e – g): Relation between AMOC strength and DSL by region for the LR-CESM, HR-POP and LR-POP. The AMOC strength and DSL by region are displayed as their differences to the first 50 years and are shown for 25-year windows, the star marker indicates the window of the onset of the AMOC collapse. A linear fit is determined through these 25-year windows, starting from the first window up to the window with the star marker, and are given in the legend ( $\phi$ ).

(Figure 3). The window length is motivated by the AMOC collapse timescale in the LR-CESM and the fact that hosing effects on DSL are expected to be relatively small over this timescale ( $\Delta F_H = 0.03$  Sv), which we will make more explicit


in Section 3.2. Relatively large DSL trends are found over the North Atlantic Ocean and Arctic Ocean during the AMOC collapse (middle row in Figure 3), with maximum DSL trends reaching +6 mm yr<sup>-1</sup>. There are also relatively large DSL trends over the Gulf Stream (extension) region which are connected to changes in the Gulf Stream path (van Westen et al., 2025a). In contrast, there are hardly any DSL trends before and after the AMOC collapse (upper and lower rows in Figure 3, respectively), indicating an acceleration in the DSL rise over the Atlantic sector during the AMOC collapse. The only exception is the HR-POP, which shows DSL trends over the last 101 model years (Figure 3h) as the AMOC is still adjusting over this period; DSL trends become smaller towards the end of the simulation (Figure 2c).

Figure 3. DSL trends over varying 101-year windows for the LR-CESM (left), HR-POP (center) and LR-POP (right), where markers indicate non-significant ( $p \ge 0.05$ ) DSL trends. The 101-year windows are (upper row): before the AMOC collapse, (middle row): during the AMOC collapse, and (lower row): after the AMOC collapse (end of simulation).

The collapsing AMOC causes DSL changes along coastal zones in the North Atlantic basin. To quantify these coastal DSL changes, we consider two densely-populated coastal zones in the western part and eastern part of the North Atlantic Ocean. First, the eastern North American coastline starting from Florida and moving northward (Figures 4a,b,c), where DSL trends during the AMOC collapse and DSL differences (between last and first 50 model years) are shown. DSL is increasing along the North American coastline during the AMOC collapse in all simulations, with DSL trends varying between +1 to +4 mm yr<sup>-1</sup>. The HR-POP shows fairly constant DSL trends with latitude, whereas there are latitudinal variations in the LR-POP (and LR-CESM). This difference between the HR-POP and LR-POP could be related to the horizontal resolution used, as the local DSL is influenced by the Florida Current through geostrophic balance (Levermann et al., 2005). To realistically resolve the


Florida Current and its responses, a high-resolution ( $\leq 0.1^{\circ}$ ) ocean component is required (Small et al., 2014). DSL differences between the last and first 50 model years greatly vary among the LR-CESM, HR-POP and LR-POP. These intermodel DSL differences are attributed to AMOC strength changes (Figures 2e,f,g), climate feedbacks, horizontal ocean resolution, and the contribution of hosing, the latter will be quantified in Section 3.2 for the LR-CESM.

Figure 4. (a – c): DSL changes along the eastern North American coastline (i.e., ocean grid cells closest to the coast) for the LR-CESM, HR-POP and LR-POP, displaying DSL trends during the AMOC collapse (black curve) and DSL differences between the last and first 50 model years (blue curve). The 101-year windows for the DSL trends are model years 1750 - 1850 (LR-CESM), 420 - 520 (HR-POP), and 1050 - 1150 (LR-POP), the spatial patterns were shown in Figures 3d,e,f. Six different coastal cities are indicated with their resident population (based on the 2020 Census). (d – f): DSL trend and DSL over the North Sea region (see inset panel d) for the LR-CESM, HR-POP and LR-POP. The DSL trends are determined over 101-year sliding windows. The dashed gray line indicates the onset of the AMOC collapse.

Second, we examine the spatially-averaged DSL changes over the North Sea region (see inset in Figure 4d), which is located in the eastern part of the North Atlantic Ocean. This semi-enclosed basin is a relatively shallow sea, with an average depth of about 100 m, and its northern boundary and southwestern boundary are connected to the North Atlantic Ocean. Sea-level variations are caused by local and remote drivers here (Dangendorf et al., 2014; Hermans et al., 2020). In the LR-CESM and LR-POP, the North Sea region is represented by only 170 grid points ( $\approx 50$  km horizontal resolution), whereas the HR-POP includes significantly more grid points, totalling to 9,784 grid points ( $\approx 7.5$  km horizontal resolution). The surface of the North Sea region receives the compensating (i.e., negative) freshwater flux forcing, but DSL does rise in all the simulations (blue curves in Figures 4d,e,f) as the background circulation transports the imposed freshwater anomalies (between 20°N to




 $50^{\circ}$ N) into the North Sea region. There is a substantial acceleration in DSL rise during the AMOC collapse (black curves in Figures 4d,e,f), with DSL trends reaching +4 mm yr<sup>-1</sup>, demonstrating that the DSL over the North Sea is strongly influenced by the AMOC.

In summary, this section presented DSL changes under a collapsing AMOC in the LR-CESM, HR-POP and LR-POP. DSLs over the North Atlantic Ocean and Arctic Ocean are influenced most under an AMOC collapse. Note that the DSL changes between the start and end of the simulations have a (substantial) hosing contribution, which will now be discussed in the section below.

#### 3.2 The hosing-corrected dynamic sea level responses

The slowly increasing freshwater flux forcing triggers the AMOC tipping event and a weaker AMOC causes DSL redistribution (Section 3.1). One unintended effect of the hosing is that it induces DSL changes through density variations. To isolate the 'pure' AMOC-induced DSL changes, we analyse the accompanying backward quasi-equilibrium LR-CESM simulation (Figure 5a); this backward simulation was not performed for the HR-POP and LR-POP (see Methods). When lowering the freshwater flux forcing, the AMOC starts to recover from model year 4,090 ( $F_H = 0.093$  Sv) and onwards, resulting in a multistable AMOC regime between  $F_H = 0.093$  Sv to  $F_H = 0.527$  Sv. DSL trends during the AMOC recovery (model year 4,090 to 4,190, Figure A1) are opposite to the ones during the AMOC collapse, the AMOC recovery results are not further discussed here. To remove the hosing contribution to DSL, one needs to subtract the different oceanic states (i.e., 'AMOC off' minus 'AMOC on') for the same  $F_H$  in the multi-stable AMOC regime.

The spatially-averaged DSL over the five regions in the full QE LR-CESM are presented in Figure 5b, which also display hysteresis behaviour. We first consider the North Atlantic Ocean region, the region that receives the hosing between 20°N to 50°N. As was argued in Section 3.1, ocean dynamics under the AMOC collapse induce DSL rise over the North Atlantic Ocean. In the backward QE simulation, the AMOC strength remains 0 Sv between model year 2,200 to 3,200 ( $\Delta F_H = 0.3$  Sv, Figure 5a) and we therefore assume that the contribution of ocean dynamics on DSL remains constant over this period. Hence, the North Atlantic DSL decline of 19.3 cm is attributed to decreasing  $F_H$  over this period (dashed red curve in Figure 5b), resulting in a DSL sensitivity of 64 cm per Sv hosing. The total hosing contribution to North Atlantic DSL then yields 42 cm  $(\Delta F_H=0.66~{
m Sv})$  and accounts for 70% of the 60 cm of North Atlantic DSL rise by model year 2,200 (at  $F_H=0.66~{
m Sv}$ ). The remaining 20 cm is attributed to different AMOC regimes, which roughly corresponds to the North Atlantic DSL differences for the same  $F_H$  in the multi-stable AMOC regime (compare the red solid and dashed curves in Figure 5b). Conversely, the hosing contribution to North Atlantic DSL during the AMOC collapse ( $\Delta F_H = 0.03$  Sv, Figures 2b and 3d) is quite small (1.9 cm), confirming our earlier assumption that DSL changes during the AMOC collapse are primarily caused internal ocean dynamics. The DSL sensitivities for the remaining regions (following the same procedure as for the North Atlantic Ocean) are: -15 cm per Sv hosing (Southern Ocean), -15 cm per Sv hosing (Indo-Pacific Ocean), 44 cm per Sv hosing (South Atlantic Ocean), and 23 cm per Sv hosing (Arctic Ocean). These sensitivities demonstrate that the North Atlantic Ocean is most sensitive under varying  $F_H$ , which is expected as the hosing is directly applied over this region.

Figure 5. (a): The AMOC strength at 1,000 m and 26°N for the forward (black curve) and backward (red curve) quasi-equilibrium LR-CESM. Markers indicate the statistical equilibria (i.e., steady states) for  $\overline{F_H}=0.18$  Sv and  $\overline{F_H}=0.45$  Sv, including error bars for their minimum and maximum values. (b): Spatially-averaged DSL differences (compared to the first 50 model years) over the five different regions (cf. Figure 2a), where solid (dashed) curves indicates the forward (backward) quasi-equilibrium LR-CESM. The time series are smoothed through a 25-year running mean to reduce the variability. (c & d): DSL differences between the statistical equilibria for  $\overline{F_H}=0.18$  Sv and  $\overline{F_H}=0.45$  Sv, displayed as the 'AMOC off' state minus the 'AMOC on' state. The markers indicate non-significant ( $p \ge 0.05$ ) DSL differences.

The spatial DSL patterns between 'AMOC off' minus 'AMOC on' are presented in Figure 5c and Figure 5d for  $\overline{F_H}=0.18$  Sv and  $\overline{F_H}=0.45$  Sv, respectively. These DSL changes are corrected for the hosing contribution and their overall patterns and amplitudes are quite similar between  $\overline{F_H}=0.18$  Sv and  $\overline{F_H}=0.45$  Sv. There are, however, some notable differences over the North Atlantic subtropical gyre, which show declining DSLs for  $\overline{F_H}=0.18$  Sv and increasing DSLs for  $\overline{F_H}=0.45$  Sv. These differences are likely not related to AMOC strength variations, as  $\overline{F_H}=0.18$  Sv and  $\overline{F_H}=0.45$  Sv have comparable AMOC strength differences ('AMOC on' minus 'AMOC off') of 11.5 Sv and 12.3 Sv, respectively (Figure 5a). DSL differences over the North Atlantic subtropical gyre can be explained by the sea surface salinity changes there; sea surface temperature responses are quite similar (Figure A2). Sea surface salinities over the subtropical gyre are increasing (i.e., lower DSLs)





between 'AMOC off' and 'AMOC on' for  $\overline{F_H}=0.18$  Sv, while decreasing (i.e., higher DSLs) for  $\overline{F_H}=0.45$  Sv. There is also salinity accumulation over the subtropical gyre at subsurface depths (250 – 500 m) for lower values of  $F_H$  and collapsed AMOC state (van Westen and Dijkstra, 2023). The different North Atlantic salinity responses between  $\overline{F_H}=0.18$  Sv and  $\overline{F_H}=0.45$  Sv can be linked to the overturning circulation in the 'AMOC off' state. There is a weak and shallow (

Figure 6. (a & b): DSL differences along the North American coastline (i.e., ocean grid cells closest to the coast) for the statistical equilibria of  $\overline{F_H} = 0.18$  Sv and  $\overline{F_H} = 0.45$  Sv for the LR-CESM, displayed as 'AMOC off' minus 'AMOC on'. Six different coastal cities are indicated with their resident population (based on the 2020 Census). (c & d): DSL climatology over the North Sea (see inset panel c) for the statistical equilibria of  $\overline{F_H} = 0.18$  Sv and  $\overline{F_H} = 0.45$  Sv for the LR-CESM. The shading indicates the 5% and 95% percentiles, the dashed lines are time-mean DSLs and are indicated in the legend.

over the North Atlantic Ocean and Southern Ocean, as there are no shared isopcynals between the regions (van Westen et al., 2025d), and is also reflected in deeper mixed layer depths over the Southern Ocean (van Westen et al., 2024b). These oceanic responses could then influence oceanic heat uptake and storage and, from this, thermosteric sea-level changes.

The  $\eta_s^g$  is decomposed into steric sea-level contributions for the five different regions and for the LR-CESM, HR-POP, and LR-POP (Figures 7b,c,d). Do note that both temperature (Figure A3) and salinity (Figure A4) changes influence the regional steric sea-level responses. After the onset of the AMOC collapse, steric sea levels are increasing over all five regions for both the LR-CESM (Figure 7b) and HR-POP (Figure 7c). However for the LR-POP (Figure 7d), only the South Atlantic Ocean, North Atlantic Ocean, and Arctic Ocean are rising after the onset of the AMOC collapse. Steric sea-levels for the South

Figure 7. (a): The global mean thermosteric sea-level rise ( $\eta_s^g$ , compared to first 50 model years) for the LR-CESM, HR-POP and LR-POP. (b – d): The steric sea-level changes (compared to first 50 model years) over the five different regions for the b) LR-CESM, c) HR-POP, and d) LR-POP. The dashed gray line indicates the onset of the AMOC collapse. The percentages in the legend indicate the fraction of the total ocean volume (with (semi-)enclosed seas and lakes only accounting for 0.06%).

Atlantic Ocean and North Atlantic Ocean slightly drop after model year 1300, which appear to be related to the development of a reversed AMOC (red curve in Figure 1a). Steric sea levels over the Indo-Pacific Ocean and Southern Ocean drop after the AMOC collapse in LR-POP, which explain the different  $\eta_s^g$  trajectory between the LR-POP with that of the LR-CESM and HR-POP (Figure 7a).

Changes in  $\eta_s^g$  are related to net heat exchange with the atmosphere. The globally-averaged surface heat flux is shown in Figures 8a,b,c (purple curves) for the LR-CESM, HR-POP and LR-POP, which is initially close to zero meaning that the ocean is in near equilibrium. The surface heat fluxes over the five regions are also displayed in Figures 8a,b,c, do note that horizontal heat exchange between the regions also influences oceanic temperatures. For example for the LR-CESM, the surface heat flux over the Indo-Pacific Ocean remains fairly constant (yellow curve in Figure 8a) while its volume-averaged temperature is



Increasing (yellow curve in Figure A3a), meaning that there is net horizontal convergence of heat into the Indo-Pacific Ocean. The most striking difference is found for the North Atlantic Ocean (red curves in Figures 8a,b,c), which initially loses heat and, after the AMOC collapse, gains heat from the atmosphere. The intermodel surface heat flux changes are also comparable for the South Atlantic Ocean (blue curves, less heat uptake), Arctic Ocean (cyan curves, less heat loss), and Indo-Pacific Ocean (yellow curves, remains fairly constant). The Southern Ocean (black curve) loses more heat after the AMOC collapse in all simulations, with the LR-POP displaying much larger responses ( $\approx$  factor of 5) compared to the LR-CESM and HR-POP. The spatial patterns in surface heat flux differences are indeed quite similar (Figures 8d,e,f), with the exception of the Southern Ocean in the LR-POP. These Southern Ocean surface heat flux responses in LR-POP highlight again differences with the LR-CESM and HR-POP, which do contribute to intermodel  $\eta_s^g$  differences (Figure 7a).

Figure 8. (a – c): The surface heat flux over the different regions and the global average (multiplied by factor 10) for the LR-CESM, HR-POP and LR-POP. The time series are smoothed through a 25-year running mean to reduce the variability. The dashed gray line in all the panels indicates the onset of the AMOC collapse. (d – f): The surface heat flux difference (last 50 minus first 50 model years) for the LR-CESM, HR-POP and LR-POP. The markers indicate non-significant ( $p \ge 0.05$ ) surface heat flux differences.

The surface heat flux responses over the Southern Ocean are quite different when comparing the HR-POP (Figure 8e) and LR-POP (Figure 8f), while both simulations have the same prescribed atmosphere. This difference is attributed to ocean eddies, which are crucial for the Southern Ocean momentum balance and oceanic responses (Stewart and Hogg, 2017; van Westen and Dijkstra, 2021). The HR-POP shows both positive and negative anomalies in sea surface temperature and mixed layer depth over the Southern Ocean (see Figure 4 in van Westen et al. (2025a)), while the LR-POP only shows increasing sea surface temperatures and mixed layer depths (see Figure S6 in van Westen et al. (2025a)). The sea surface temperature responses eventually control the sign of surface heat flux changes, as higher (lower) sea surface temperatures increase (decrease) the





temperature difference with the overhead atmosphere and result in greater (smaller) heat loss over the Southern Ocean. The interaction between ocean eddies with the Antarctic Circumpolar Current also induces a mode of Southern Ocean multidecadal variability (40 – 50 years) that propagates through the global ocean (Le Bars et al., 2016; van Westen and Dijkstra, 2017) and is visible in the HR-POP time series (e.g., Figures 7a,c and Figure 8b). A limitation of both HR-POP and LR-POP is their prescribed atmosphere, which effectively implies an infinite atmospheric heat capacity (Le Bars et al., 2016). Hence, the LR-CESM needs to be analysed to consider the energy balance of the entire climate system.

The total energy budget of system Earth can be quantified by analysing the radiative imbalance at the top of atmosphere (TOA) in the LR-CESM. The responses in the radiative imbalance at TOA (Figure 9a) closely resemble those of the surface heat flux (purple curve in Figure 8a), indicating that the ocean primary stores (releases) the net incoming (outgoing) energy at TOA. This is also evident from the much larger oceanic heat capacity than the atmospheric heat capacity (Von Schuckmann et al., 2020). The slightly negative radiative imbalance at TOA results in a 7.5 cm drop in  $\eta_s^g$  prior to the AMOC collapse (Figure 7a). Thereafter, the radiative imbalance becomes positive and  $\eta_s^g$  increases by 21 cm between model years 1,758 and 2,200. Over the same period, the mass-weighted atmospheric temperature drops by 0.19°C (not shown), confirming that the net energy input at TOA is primarily stored in the ocean.

To further understand the abrupt increase in energy input at TOA during the onset of the AMOC collapse, we decompose the TOA radiative imbalance into its incoming shortwave radiation (SW<sup>in</sup>, Figure 9b) contribution and outgoing longwave radiation (LW<sup>out</sup>, Figure 9c) contribution. Under an AMOC collapse, both the globally-averaged SW<sup>in</sup> and LW<sup>out</sup> at the TOA decline, although inter-hemispheric differences remain. The SW<sup>in</sup> decreases over the Northern Hemisphere by the greater sea-ice cover and the opposite is true for the Southern Hemisphere, but the Northern Hemispheric sea-ice response dominates and there is a net increase in the planetary albedo (van Westen et al., 2024b). Consequently, the Northern Hemisphere cools and emits less longwave radiation (i.e., the Planck feedback) and again the opposite is true for the Southern Hemisphere. The globally-averaged response in LW<sup>out</sup> is slightly stronger than in SW<sup>in</sup>, resulting in the positive radiative imbalance at TOA during and after the AMOC collapse. Note that there are also regional climate feedbacks that alter the local radiative imbalance, such as the southward migration of the Intertropical Convergence Zone (ITCZ) that is visible in both SW<sup>in</sup> and LW<sup>out</sup> components (Figures 9e.f), but not so much in the radiative imbalance (Figures 9d).

A collapsing AMOC affects DSLs (Section 3), steric sea levels, surface heat fluxes and Earth's energy imbalance. The AMOC-induced changes are dependent on horizontal resolution used (strongly eddying versus eddy parametrisation) and configuration used (coupled versus stand-alone ocean). In the coupled simulation (LR-CESM), the findings presented in this section demonstrate that an AMOC collapse leads to a substantial thermosteric sea-level rise (> 20 cm), driven by increased oceanic heat uptake from a positive radiative imbalance at the TOA. This does not contradict the findings by Gregory et al. (2024) and Vogt et al. (2025), where they argue that the AMOC strength and oceanic heat uptake are not related. Indeed, when the AMOC reduces to zero in the LR-CESM, it effectively halts downwelling of heat in the Atlantic Ocean. Net oceanic heat uptake is ultimately stored in regions outside the Atlantic Ocean (Figure A3a), underscoring the key role of the AMOC in modulating Earth's energy balance.

Figure 9. (a − c): The globally-averaged radiative imbalance at the top of atmosphere (panel a) for the LR-CESM. The inset shows the radiative imbalance difference compared to first 50 model years, which is also split for different latitude bands. The radiative imbalance is decomposed into an incoming shortwave radiation (SW<sup>in</sup>, panel b) contribution and outgoing longwave radiation (LW<sup>out</sup>, panel c) contribution. The SW<sup>in</sup> and LW<sup>out</sup> time series are displayed as differences (compared to the first 50 model years) and for different latitude bands. All time series are smoothed through a 25-running mean to reduce the variability. The dashed gray line indicates the onset of the AMOC collapse. (d – f): The radiative imbalance at top of atmosphere, SW<sup>in</sup> and LW<sup>out</sup> differences for model years 2151 – 2200 (compared to first 50 model years). The markers indicate non-significant ( $p \ge 0.05$ ) differences.

#### 5 Conclusions




We presented results from the fully-coupled climate model (LR-CESM) and a high-resolution and low-resolution stand-alone ocean model (HR-POP and LR-POP), which were forced under a slowly increasing freshwater flux forcing (van Westen et al., 2024b, 2025a). Our aim was to revisit AMOC-induced DSL responses (Levermann et al., 2005) by analysing DSL changes arising from intrinsic ocean dynamics that lead to an AMOC collapse.

The regional DSL is controlled by ocean density and ocean dynamics. We have demonstrated that ocean dynamics strongly control DSLs over the North Atlantic Ocean and Arctic Ocean. An AMOC collapse induces the largest DSL rise over these two regions, with DSL trends reaching 6 mm yr<sup>-1</sup> over a 100-year period. Both the collapsing AMOC dynamics and the hosing contribute to these positive DSL trends, although the latter contribution is relatively small. DSL trends of 6 mm yr<sup>-1</sup> represent a considerable local increase, given that the average global mean sea-level rise was 3.3 mm yr<sup>-1</sup> over the period 1993 – 2024 (Hamlington et al., 2024). Coastal regions along the North Atlantic Ocean could see an acceleration in DSL rise under an AMOC collapse, although the results are dependent on the model configuration used (coupled versus stand-alone ocean) and the horizontal ocean resolution used (strongly eddying and eddy parameterisation). Apart from ocean dynamic changes, the AMOC also influences the globally-averaged radiative imbalance and this modifies the oceanic heat content. A collapsed



AMOC causes a positive radiative imbalance at top of atmosphere (maximum of +0.5 W m<sup>-2</sup>) and results in greater oceanic heat uptake leading to global mean thermosteric sea-level rise of more than 20 cm. The resulting sterodynamic sea-level change (i.e.,  $\eta_s^g$  + DSL; not shown) becomes positive almost everywhere under a substantially weakened AMOC.

The presented DSL trends and changes in Section 3.1 do have an unintended hosing contribution in the LR-CESM, HR-POP and LR-POP, where DSL changes over the North Atlantic Ocean and Arctic Ocean are substantially overestimated. For relatively small changes in the hosing forcing, such as the 100-year window during the AMOC collapse ( $\Delta F_H = 0.03$  Sv), intrinsic ocean dynamics dominate and DSL trends and changes are not affected much by the imposed hosing. For larger hosing intervals (e.g., end of simulation minus begin of simulation), this hosing contribution needs to be considered. To remove the hosing contribution to DSL changes, the accompanying backward QE LR-CESM simulation was used (van Westen and Dijkstra, 2023). The hosing-corrected DSL changes are obtained by considering the different oceanic regimes ('AMOC on' and 'AMOC off') within the multi-stable AMOC regime and for the same hosing forcing, which are then the AMOC-induced DSL responses. For example, for the North Sea region, the hosing-corrected DSL change between 'AMOC off' and 'AMOC on' is about 50 cm, with the hosing contribution ( $\Delta F_H = 0.66$  Sv) adding a further 30 cm of DSL rise. Thus, DSL changes should be considered with care in the presence of a varying freshwater flux forcing.

The LR-CESM has constant pre-industrial greenhouse gas concentrations, which allows to nicely isolate DSL due to just the AMOC collapse. For assessing impacts, it is also relevant to study DSL responses under climate change (Ferrero et al., 2021; Pardaens, 2023), as the overall impact on DSL depends on greenhouse gas emission scenario and timing of AMOC collapse. van Westen et al. (2025d) recently performed such climate change simulations, in which the LR-CESM was forced under an intermediate-emission scenario and a high-emission scenario. The drawback of these climate change simulations is that they were performed under constant freshwater flux forcings of  $\overline{F_H} = 0.18$  Sv and  $\overline{F_H} = 0.45$  Sv, which may influence the DSL responses. A few CMIP6 simulations are available that exhibit a collapsing AMOC under climate change (Drijfhout et al., 2025) and are well suited for analysing DSL responses. Future work will address these DSL changes in LR-CESM and different CMIP6 models under climate change.

The global mean sea level is projected to increase in the upcoming decades to centuries under future climate change (Turner et al., 2023) and an acceleration in the global mean sea-level rise poses challenges for successful adaptation strategies to sealevel rise (Haasnoot et al., 2018; Hamlington et al., 2024). An AMOC collapse could exacerbate local sea-level rise projections in the North Atlantic Ocean. Densely-populated coastal regions along the North Atlantic may experience DSL trends of up to 4 mm yr<sup>-1</sup> and a total DSL rise of 50 cm when the AMOC has fully equilibrated to its collapsed state. The currently observed sea-level rise of 3 mm yr<sup>-1</sup> over the North Sea (Steffelbauer et al., 2022; Keizer et al., 2023) would then double if the AMOC collapses. It is therefore important that future sea-level rise projections for the North Atlantic Ocean consider the effects of an AMOC collapse scenario (Biesbroek et al., 2025).

Code and data availability. All model output and code to generate the results are available at: https://doi.org/10.5281/zenodo.17285384

Figure A1. DSL trends over 101-year windows for the backward quasi-equilibrium LR-CESM, where markers indicate non-significant ( $p \ge 0.05$ ) DSL trends. The 101-year windows are (a): before the AMOC recovery, (b): during the AMOC recovery, and (c): after the AMOC recovery (the last 101 model years).

Figure A2. (a & b): Sea surface temperature differences between the statistical equilibria for  $\overline{F_H}=0.18$  Sv and  $\overline{F_H}=0.45$  Sv, displayed as the 'AMOC off' state minus the 'AMOC on' state. (c & d): Sea surface salinity differences between the statistical equilibria for  $\overline{F_H}=0.18$  Sv and  $\overline{F_H}=0.45$  Sv, displayed as the 'AMOC off' state minus the 'AMOC on' state. The markers indicate non-significant ( $p \ge 0.05$ ) differences.

Author contributions. R.M.v.W., C.A.K and D.L.B. conceived the idea for this study. R.M.v.W. conducted the analysis and prepared all figures. All authors were actively involved in the interpretation of the analysis results and the writing process.

Competing interests. The authors declare no competing interests.


**Figure A3.** Volume-averaged oceanic temperature responses for the five different regions and global mean for the LR-CESM (left), HR-POP (middle) and LR-POP (right). The full-depth temperatures (upper row) are decomposed into an upper 1,000 m contribution (middle row) and below 1,000 m contribution (lower row). All time series are displayed as differences compared to the first 50 model years and are smoothed through a 25-running mean to reduce the variability. The dashed gray line indicates the onset of the AMOC collapse.

Acknowledgements. The model simulations and the analysis of all the model output was conducted on the Dutch National Supercomputer (Snellius) within NWO-SURF project 2024.013 (PI: Dijkstra). All the model output was generated as part of the ERC-AdG project TAOC (project 101055096; PI: Dijkstra).

**Figure A4.** Similar to Figure A3, but now for the salinity.

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
