# Peer review of "Dynamic and Steric Sea-level Changes due to a Collapsing AMOC in the Community Earth System Model"

_EGUsphere, 2025_

## Author Comment (AC1)

**MS-No.:** egusphere-2025-5102

**Version:** Revision

**Title:** Dynamic and Steric Sea-level Changes due to a Collapsing AMOC in the Community Earth System Model

**Author(s):** René M. van Westen, Caroline A. Katsman and Dewi Le Bars

**Point-by-point reply to reviewer**

January 16, 2026

We thank the reviewer for their careful reading and for the useful comments on the manuscript.

*The study by van Westen et al. investigates dynamic and steric sea-level responses to an AMOC collapse using hosing experiments conducted with a fully coupled model and two ocean-only models with different horizontal resolutions. The authors find that an AMOC collapse induces dynamic sea-level rise across the Atlantic and Arctic Oceans, with the largest signals in the North Atlantic. Meanwhile, the collapse produces global-mean thermosteric sea-level rise due to enhanced net downward top-of-atmosphere (TOA) energy flux, most of which is absorbed by the ocean through increased ocean heat uptake. Overall, the results are robust and clearly presented. I recommend minor revision with the following comments.*

1. *Line 22: The term 'regional' needs clarification. Does this refer specifically to the Atlantic sector? Similarly, does 'background ocean circulation' correspond to the AMOC, or does it include gyre circulations or other components? Please specify to avoid ambiguity.*

   **Author's reply:**

   We agree that both 'regional' and 'background' are confusing here. This needs to be clarified or removed.

   **Changes in manuscript:**

   We will check the entire manuscript on this and change the text accordingly.

2. *Line 91: The details of the prescribed surface forcing in the ocean-only simulations should be clarified. My interpretation is that momentum fluxes are prescribed, while heat fluxes are interactive.*

   **Author's reply:**

   We agree that this needs to be clarified. In the POP simulations, the momentum fluxes (i.e., wind stress) are prescribed, while heat and evaporative fluxes are interactive through sea surface temperatures.

   **Changes in manuscript:**

   We will clarify the POP set-up and rewrite the Methods.

3. *Line 277-290: Related to the previous comment: Are surface heat fluxes prescribed in the POP simulations? If they are interactive, then surface-flux feedbacks may also contribute to the large ocean heat-uptake anomalies in the North Atlantic and Southern Ocean. This could partially explain the spatial heterogeneity of heat-uptake responses across the models.*

   **Author's reply:**

   The surface heat fluxes are interactive, only the near-surface atmospheric temperatures are prescribed. We agree with the reviewer that we need to clarify the POP set-up in greater detail, which helps to understand the heat-uptake responses.

   **Changes in manuscript:**

   We will clarify this in the Methods (see also point 2 by the reviewer).

4. *Figure 4b: The dynamic sea-level (DSL) response shows a positive trend in the time series but exhibits negative anomalies north of 30°N in the spatial difference map. How these two diagnostics relate?*

   **Author's reply:**

   Note that DSL trends use the left vertical axis, while DSL differences the right vertical axis. To avoid confusion, we will add this in the caption and make this more explicit in the main text.

**Changes in manuscript:**

We will rewrite parts of the caption and main text.

5. *Figure 8: The Southern Ocean heat-uptake response differs markedly between HR-POP and LR-POP, consistent with the role of mesoscale eddies highlighted in the manuscript. However, the physical teleconnection between AMOC collapse (or hosing) and Southern Ocean eddy activity is not discussed. A brief explanation of the mechanism, e.g., changes in wind stress, ACC baroclinicity, or remote propagation of density anomalies would strengthen the interpretation.*

   **Author's reply:**

   The eddy activity is modified under a different Southern Ocean stratification following the AMOC collapse, which is being addressed in a follow-up study (manuscript submitted to OS, Smolders et al.). We can expand the discussion in lines 258 – 267.

   **Changes in manuscript:**

   We will expand the discussion here and add a reference to the preprint.

6. *How many ensemble members are conducted in each experiment? Will the internal variability affect the results such as AMOC tipping time?*

   **Author's reply:**

   One realisation is available for each experiment due to computational constraints. We agree with the reviewer that this need to be explicitly mentioned in the manuscript. Internal variability may influence the AMOC tipping time (e.g., Romanou et al., 2023; https://doi.org/10.1175/JCLI-D-22-0536.1), but the destabilising feedback operates independently of internal variability once the saddle-node bifurcation is crossed. Comparable AMOC collapse trajectories have been found in the LR-CESM (https://doi.org/10.5194/esd-16-2063-2025, their Figure 2). It is good to briefly mention this in the paper.

   **Changes in manuscript:**

   In the Methods, we will include that one realisation is available for each experiment and discuss the role of internal variability.

1. *Line 129: how are the basin boundaries are defined?*

   **Author's reply:**

   It is good to elaborate on the choices of the basin boundaries. The boundaries are primarly based on the meridional extent of the AMOC, the AMOC is dominant from 34°S to 65°N. The ocean surfaces south of 34°S define the Southern Ocean region and north of 65°N (up to the Bering Strait) define the Arctic Ocean region. The remaining ocean surfaces are attributed to the Indo-Pacific Ocean region.

   **Changes in manuscript:**

   We will add a description of the basin boundaries in the revision.

2. *Line 129: determined - determine*

   **Author's reply:**

   Agreed.

   **Changes in manuscript:**

   Will be corrected.

---

## Author Comment (AC2)

**MS-No.:** egusphere-2025-5102

**Version:** Revision

**Title:** Dynamic and Steric Sea-level Changes due to a Collapsing AMOC in the Community Earth System Model

**Author(s):** René M. van Westen, Caroline A. Katsman and Dewi Le Bars

**Point-by-point reply to reviewer**

January 16, 2026

We thank the reviewer for their careful reading and for the useful comments on the manuscript.

*van Westen et al. present an analysis of sea-level changes resulting from an AMOC collapse using a set of numerical experiments with the Community Earth System Model in which a freshwater hosing is performed over the North Atlantic regions to induce an AMOC collapse. The experiments used form an interesting model hierarchy to study sea-level changes with both low- and high-resolution ocean-only simulations, and a single low-resolution fully coupled simulation. Given the lack of previous studies, especially using modern ocean models, I think the results of the present study are timely. I think the manuscript is generally written to a good standard, structured appropriately, and the results presented with an appropriate degree of clarity. However, I do think the DSL analyses, and some of the claims made from these are limited by the absence of a reverse hosing simulation in the ocean-only configurations. Nevertheless, given the structure of the model hierarchy used, there is potential to better explain the mechanisms and how these compare between the different experiments of the present study. This could be important for interpreting DSL/TS sea level changes in other climate models. Lastly, I think computing the total sterodynamic sea level change would be helpful – currently it is hard to determine how significant the overall changes are. Ultimately, I suggest moderate revisions before publication in Ocean Sciences.*

Specific comments:

1. *L8-11: I think it's good to include this information in the abstract, however, could the authors quote/estimate these values in terms of*

*mm/year sea-level rise? This would be helpful for readers to make a quick comparison to the overall DSL changes.*

**Author's reply:**

We agree with the reviewer that is good to have a reference for the AMOC-induced DSL changes. The observed global mean sea-level rise of $+3.3$ mm yr$^{-1}$ is well suited for such a reference. We are aware that local sea-level rise may deviate from the global mean and sea-level rise is also forcing dependent. The latter can be included in the discussion, but is too specific for the abstract.

**Changes in manuscript:**

We will rewrite parts of the abstract and discussion accordingly.

2. *L16: Somewhere in this paragraph I think it would be appropriate to cite:*

   *Liu et al. Sci. Adv. (https://doi.org/10.1126/sciadv.aaz4876)*

   *Bellomo and Mehling GRL. (https://doi.org/10.1029/2023GL107624)*

   **Author's reply:**

   These are indeed relevant references here.

   **Changes in manuscript:**

   These references will be incorporated in this paragraph.

3. *L35: Could also cite Baker et al. 2025, Nature (already cited in the manuscript).*

   **Author's reply:**

   Agreed.

   **Changes in manuscript:**

   The study will be mentioned here.

4. *L55-57: I think it would be better to split the Methods section more cleanly between each model used (e.g., with subheadings) and provide a little more information on the simulation components to make this a*

*more standalone manuscript. I understand these experiments have been used in several previous studies, but for completeness in the current paper I think the authors should at least briefly state which components of CESM are being used here (e.g., is this FV/spectral CAM; CICE? etc.). A short table would help.*

**Author's reply:**

Agreed, we can expand on the descriptions of the LR-CESM and the two stand-alone POP simulations. We will introduce the LR-CESM in Section 2.1, together with information on the different CESM components. Section 2.2 describes the two stand-alone POP configurations in more detail.

**Changes in manuscript:**

We will rewrite the Methods with this comment in mind (Section 2).

5. *L98: The authors need to expand on this more. Critically, are there reasons to think that the hosing effect would be significantly different in these experiments? One of the main takeaways from the analysis performed in section 3.2 is that the hosing corrected, AMOC-induced DSL change is highly dependent on the AMOC state and hosing conditions, but this has only been explicitly calculated for one simulation. These effects could be highly dependent on the model resolution.*

**Author's reply:**

We agree with the reviewer that the hosing-corrected DSL changes are highly dependent on the model (resolution) used and the AMOC state, which was demonstrated for the LR-CESM in Section 3.2. Unfortunately, this cannot be tested for the HR-POP and LR-POP due to computational limitations (lines 97 – 98), which is also discussed in the last section (lines 349 – 359). Nevertheless, this discussion can be expanded to address the opportunities and limitations of the analysed model output.

**Changes in manuscript:**

We will expand the discussion with regard to this topic.

6. *L146-159/Figure 2: The authors have not justified why they are using a linear fit for dynamic sea level difference and 'Volume transport difference' in figures 2.e-g. It's potentially interesting if there is a well-justified physical relationship which holds over some region of the parameter space, but I think this should be far better justified/explained if it is to be included.*

**Author's reply:**

The AMOC-DSL relation was already briefly described in lines 147 – 148, but this can be made more explicit following the reviewer's suggestion. The AMOC is destabilised under the salt-advection feedback, leading to freshwater accumulation over the North Atlantic Ocean. This freshwater accumulation increases local DSLs (through lower densities), explaining the linear AMOC-DSL relation.

**Changes in manuscript:**

We will motivate the AMOC-DSL relation in greater detail.

7. *L152-153: I do not think this statement (as it is currently written) is justified by figure 2.*

**Author's reply:**

Agreed, this should be quantified. We will quantify the deviations from the AMOC-DSL relation.

**Changes in manuscript:**

A quantitative assessment will be added in the revision (e.g., a Table in the Appendix).

8. *L180: This needs to be substantiated. For instance, which climate feedbacks and why?*

**Author's reply:**

Agreed, we will further motivate these contributions. The LR-CESM and LR-POP only differ in their climate coupling, hence climate feedbacks (e.g., winds) are not represented in the LR-POP.

**Changes in manuscript:**

The text will be rewritten accordingly.

9. *L198: Ultimately it is up to the authors, but perhaps it would make more sense to discuss the magnitude of the hosing correction first? Or, even include this in the methods where it is first introduced? Presently, the reader is introduced to changes in DSL which initially appear very large, but the end of section 3.1 ends with an acknowledgment of the substantial hosing correction, and the correspondingly large reduction in DSL trends. Importantly, many figures show the uncorrected DSL results.*

   **Author's reply:**

   The hosing contribution to DSLs is spatially and AMOC-state dependent over the North Atlantic Ocean, whereas other regions have a negligible hosing contribution. Over 100-year windows ($\Delta F_H = 0.03$ Sv, Figure 3), the hosing contribution is relatively small and hence the DSL trends can be compared for the different models. Note that the hosing contribution to DSLs is the largest for LR-CESM ($\Delta F_H = 0.66$ Sv), which is followed by the LR-POP ($\Delta F_H = 0.45$ Sv), and is the smallest for the HR-POP ($\Delta F_H = 0.18$ Sv). Nevertheless, we agree with the reviewer that it is good to already mention this in the Methods, but we will keep the overall outline of the manuscript.

   **Changes in manuscript:**

   We will further stress the (substantial) hosing contribution to DSLs in the Methods.

10. *L255-257 Here, LR-POP and LR-CESM yield significantly different responses at the same resolution – I think there is an opportunity here to explore the role of eddies in more depth than is currently presented.*

    **Author's reply:**

    Please note that both the LR-CESM and LR-POP parameterise ocean eddy effects, the role of ocean eddies can only be analysed between the HR-POP and LR-POP. Nevertheless, the differences in the thermosteric sea-level responses are discussed in lines 268 through 303. For example in Figure 8, the surface heat flux responses are quite similar between the models, except for the Southern Ocean region in the

LR-POP. The Southern Ocean responses are much larger for the LR-POP compared to the LR-CESM and HR-POP (line 287), explaining the intermodel thermosteric sea-level differences. The role of Southern Ocean eddies is discussed in lines 291 – 301.

**Changes in manuscript:**

No changes needed.

11. *L262: Only in the adiabatic limit.*

    **Author's reply:**

    Correct, this is only valid under thermal wind balance and the adiabatic limit.

    **Changes in manuscript:**

    This will be included in the revision.

12. *L267: Ocean heat content can be easily computed from POP variables (fig. A3 partially addresses this).*

    **Author's reply:**

    Correct, the ocean heat content responses are essentially shown by the volume-averaged temperatures in Figure A3.

    **Changes in manuscript:**

    We will add a reference to Figure A3 here.

13. *Figure 7: What is the difference in thermosteric sea level rise between the forward and backward simulations?*

    **Author's reply:**

    The reviewer suggested (see reviewer's introduction) to include the sterodynamic sea-level responses, which is a relevant suggestion. The sterodynamic sea-level responses also have the unintended hosing contribution when analysing the forward QE simulations, hence we will include an analysis on the global mean thermosteric sea-level changes using the full hysteresis simulation with the LR-CESM.

**Changes in manuscript:**

We will add a new section 4.2 and present the results on sterodyanmic sea-level responses, together with the global mean thermosteric sea-level changes in the hysteresis simulation.

14. *L305-309: I think the authors need to take a lot more care here. Arguments based on a link between AMOC and TOA imbalance can only be applied to LR-CESM (as acknowledged). The surface fluxes are presumably the most important aspect - for instance, the surface fluxes in LR-CESM are not 0 prior to the AMOC collapse.*

    **Author's reply:**

    The reviewer is correct here. AMOC strength variations change the surface heat fluxes and they alter the TOA radiative imbalances.

    **Changes in manuscript:**

    We will rewrite this paragraph and other parts (abstract and discussion) of the manuscript.

15. *L340-342: How does this value compare to estimates of sea level rise due to accelerated melting of the Greenland (or Antarctic) ice sheet? How large is the overall sea level change due to AMOC collapse?*

    **Author's reply:**

    It is indeed good to compare the AMOC-induced DSL changes with the other sea-level rise contributions. The hosing simulations have constant greenhouse gas concentrations, hence we will compare the projected sea-level changes from the SSP scenarios.

    **Changes in manuscript:**

    We will rewrite these sentences accordingly.

16. *L348: I think it would be helpful to show this somewhere in the manuscript. Again on L372-373, should the total sea-level change be quoted?*

    **Author's reply:**

    The sterodynamic sea-level responses will be included in the revised manuscript (see comment 13). We will also rewrite the last paragraph (see also comment 15).

**Changes in manuscript:**

The manuscript will be changed accordingly.

Technical Points

1. *L69: "which occurs at $\Delta F_H = 0.03$ Sv" ?*

   **Author's reply:**

   This is the 100-year timescale of the AMOC collapse.

   **Changes in manuscript:**

   This will be clarified in the revised manuscript.

2. *L151: "quantity" ?*

   **Author's reply:**

   Agreed.

   **Changes in manuscript:**

   Will be corrected.

3. *L220: Better to put these values in a table?*

   **Author's reply:**

   Agreed, these values can be put in a table.

   **Changes in manuscript:**

   We will add a table in the Appendix.

4. *L253: salinity is already a concentration.*

   **Author's reply:**

   Agreed.

   **Changes in manuscript:**

   We will remove 'concentration'.

5. *L304: "of the Earth system" ?*

   **Author's reply:**

   Agreed.

   **Changes in manuscript:**

   Will be corrected.

6. *L306: "primarily" ?*

   **Author's reply:**

   Agreed.

   **Changes in manuscript:**

   Will be corrected.

7. *L360: "...which allows us to..." ?*

   **Author's reply:**

   Agreed.

   **Changes in manuscript:**

   Will be corrected.

8. *Figure 4: A lot of the text here (and in other figures) is very small – could the authors increase the size? Moreover, some of the inset panels clutter the figures.*

   **Author's reply:**

   We will increase the fontsize in panels 4a,b,c and 6a,b. We will remove the insets in panels 2b,c,d to declutter the panels, these DSL differences can be moved to the Appendix.

   **Changes in manuscript:**

   We will incorporate these changes in the revision.

9. *Figure 7: are panels b-d the total steric contribution, or just the thermosteric contribution, as shown in panel a? Are we meant to compare panel a to panels b-d?*

   **Author's reply:**

   Panels b,c,d display the total steric contribution, as is mentioned in the caption and Lines 268 – 270.

   **Changes in manuscript:**

   No changes needed.